# What Protects Youth Residential Caregivers from Burning Out? A Longitudinal Analysis of Individual Resilience

**DOI:** 10.3390/ijerph17072212

**Published:** 2020-03-25

**Authors:** Nina Kind, David Bürgin, Jörg M. Fegert, Marc Schmid

**Affiliations:** 1Child and Adolescent Psychiatry, University Basel, Psychiatric University Hospital Basel, Wilhelm Klein-Strasse 27, 4002 Basel, Switzerland; david.buergin@upkbs.ch (D.B.); marc.schmid@upkbs.ch (M.S.); 2University Hospital Ulm, Department for Child and Adolescent Psychiatry and Psychotherapy, Steinhövelstrasse 5, 89075 Ulm, Germany; Joerg.Fegert@uniklinik-ulm.de

**Keywords:** residential care, burnout, stress, resilience, sense of coherence, self-efficacy, self-care

## Abstract

*Background*: Professional caregivers are exposed to multiple stressors and have high burnout rates; however, not all individuals are equally susceptible. We investigated the association between resilience and burnout in a Swiss population of professional caregivers working in youth residential care. *Methods*: Using a prospective longitudinal study design, participants (*n* = 159; 57.9% women) reported on burnout symptoms and sense of coherence (SOC), self-efficacy and self-care at four annual sampling points. The associations of individual resilience measures and sociodemographic variables, work-related and personal stressors, and burnout symptoms were assessed. Cox proportional hazards regressions were calculated to compute hazard ratios over the course of three years. *Results*: Higher SOC, self-efficacy and self-care were related to lower burnout symptoms in work-related and personal domains. Higher SOC and self-efficacy were reported by older caregivers and by those with children. All three resilience measures were highly correlated. A combined model analysis weakened the protective effect of self-efficacy, leaving only SOC and self-care negatively associated with burnout. *Conclusion*: This longitudinal analysis suggests that SOC and self-caring behaviour in particular protect against burnout. Our findings could have implications for promoting self-care practices, as well as cultivating a meaningful, comprehensible and manageable professional climate in all facets of institutional care.

## 1. Background

Professional caregivers working in youth residential care are exposed to multiple stressors and have high burnout rates [1,2,3,4]. A Swiss national survey found 80% of children and adolescents reporting traumatic experiences [5], and the majority show clinically relevant internalizing and/or externalizing behaviour [5,6,7,8]. This vulnerable clientele, often exhibiting severely disruptive social behaviour, are supervised by professional caregivers in physically and emotionally demanding shifts around the clock.

Burnout is characterized by feelings of disempowerment, emotional exhaustion, cynicism, depersonalization, anxiety and loss of confidence [2,3,9,10]. Studies estimating the prevalence of burnout have suggested that as many as 50% of child protection workers report burnout symptoms [10,11,12]. In their meta-analyses, Mor Barak et al. reported burnout as being one of five variables with the largest standardized effect size associated with turnover in social work [13]. When work demands become overwhelming, the risk of burning out increases, which poses a problem for work satisfaction, employee turnover and quality of care [5,14,15,16]. In particular, the ineffective aspect of burnout has been largely disregarded in stress research [4]. However, not all individuals are equally susceptible to developing burnout symptoms.

Numerous studies have assessed individual resilience to such work-related stress, which the American Psychological Society defines as the process of ‘bouncing back’ and adapting in the face of difficult experiences [17,18]. One well-known measure of resilience is sense of coherence (SOC), established as an integral variable related to the professional functioning of an individual [19]. The term was coined by Aaron Antonovsky and reflects one’s perception of life as being comprehensible, manageable and meaningful. Other resilience measures include self-caring behaviour (e.g., team supervision, work-life balance, physical health, social support) and perceived self-efficacy—a subjective belief in the ability to execute the actions required to manage situations [20,21,22,23,24,25,26].

Stress research supports an association between individual resilience measures and burnout [27,28,29,30,31]. In a study on Polish social workers, a higher SOC was related to fewer burnout symptoms [32]. In their cross-sectional study with 2053 Danish employees, Albertson et al. found that people with high levels of SOC experienced less stress symptoms [33]. Feldt found that they were also more protected from adverse psychological effects of stressful work conditions [34]. Perceived self-efficacy of staff had a positive influence on burnout symptoms [35,36,37]. A meta-analysis of 57 studies by Shoji et al. even demonstrated significant self-efficacy–burnout relationships across countries and professions, while workers who engaged in more self-care reported lower burnout levels [21,22,38]. Based on such findings, individual attitudes and behaviours may reduce the likelihood of feeling threatened by adverse work conditions or less vulnerable thereafter, and more readily able to cope with future stressors.

Sociodemographic factors such as sex, younger age, shift work and being single have also been linked to increased burnout risk [39,40,41,42,43,44]. Inversely, higher SOC scores and self-efficacy were found in older age cohorts, and in those with more work experience [24,38,45].

Investigating the buffering role of individual resilience on burnout is highly relevant for youth welfare organizations and occupational health policies. Despite the broad band of resilience and burnout research, as well as the implications for job performance, organizational commitment and job dissatisfaction, healthcare costs, and staff turnover, the long-term impact in the domain of youth residential care remains largely unexplored [3,5,46,47].

Drawing on the theoretical and empirical evidence, the main aim of this study was to investigate the longitudinal association between specific resilience measures and burnout in a Swiss population of professional caregivers working in youth residential care.

## 2. Methods and Measures

We conducted this prospective study as part of a larger government-funded model project examining the efficacy of trauma-informed care in residential youth welfare institutions in the German speaking part of Switzerland. Included welfare institutions were approved by the Swiss Federal Office of Justice and incorporate a broad range of clients placed in out-of-home care for both civil reasons (e.g., family conflicts, neglect) and juvenile justice reasons (e.g., educational measures, reintegration). Six institutions were sex-specific (3 for boys, 3 for girls, ages 12–20 years old), while eight institutions were co-educative (ages 5–20 years old). The 14 youth welfare institutions offer placements for almost 300 clients. Managers and employees were trained in specific care concepts, and caregiver burden, attitudes and resilience were assessed over the course of three years. Sample size estimations were originally based on the primary study aim. Due to the secondary, explorative nature of the current analyses, no sample estimation to determine response rate was conducted.

### 2.1. Study Population

A total of 168 employees were enrolled in the study, but 9 of them were excluded due to missing data in baseline variables. Thus, 159 professional caregivers, i.e., social pedagogues or social pedagogues in training (67 men, 92 women) aged between 22 and 61 years (mean = 35.85, SD = 9.68) who worked in 14 residential youth welfare institutions were included in the study. On average, they had 8.3 years (range = 0–37) of professional experience in residential youth welfare institutions and had worked in the present institution for a mean of 3.9 years (range = 0–21). Two years of professional experience and a working history in the present institution of one year were most frequently reported.

### 2.2. Procedures

We used a longitudinal design over the course of three years to estimate changes in the reported burnout of youth residential care staff over time. Surveys and well-established questionnaires were mailed to partaking institutions at four annual sampling points between 2012 and 2015. Participants were continuously included in the study, with an average of 10.5 months between individual measurements. Not all participants had data for all four measures, since some started working in the institutions during the course of the study or missed a data collection due to absences (e.g., vacation, illness). Data were collected from surveys on sociodemographic variables, experiences of personal and work-related stressors, and self-caring behaviour, as well as questionnaires on burnout, SOC and perceived self-efficacy

### 2.3. Ethics Approval and Consent to Participate

All participants were thoroughly informed about the study, and they gave written informed consent. The leading Ethics Committee Basel-Stadt and Basel-Land (EKBB, Ref.Nr. 288/12), as well as the Cantonal Ethics Committee Bern (KEK-BE, Ref.Nr. 014/13), Ethics Committee St. Gallen (EKSG, Ref.Nr. 13/003), Ethics Committee Appenzell Ausserrhoden (EKAR, Ref.Nr. 34), Cantonal Ethics Committee Luzern (KEK-LU, Ref.Nr. 13009) and the Cantonal Ethics Committee Zürich (KEK-ZH, Ref.Nr. 2013-0030) approved this model project.

### 2.4. Measures

#### 2.4.1. Burnout Screening Scale

The Burnout Screening Scale (BOSS) is a standardized and validated questionnaire to collect information on current psychological (cognitive and emotional), somatic, and psychosocial symptoms in work-related, personal, and interpersonal domains which are related to burnout [48]. The validity of this measure was established in large samples [49,50]. The first part of the questionnaire assesses symptoms in different life domains (work, personal life, family and friends) during the last three weeks (4 subscales with 30 items). The second part of the questionnaire assesses clinical (somatic, cognitive, and emotional) symptoms during the last seven days (3 subscales with 30 items). A 6-point Likert scale scored from 1 = “does not apply” to 6 = “applies strongly” is used. According to Hagemann and Geuenich, burnout is suspected if one or more values on the 10-item work scale are elevated (T-score ≥ 60). The authors reported Cronbach‘s alpha between 0.75 and 0.91.

#### 2.4.2. Sense of Coherence Scale

The sense of coherence in regards to daily work was assessed with a well-established German short version of the Sense of Coherence Scale by Antonovsky and Franke (7-point Likert scale with 9 items, scored from 1 = “very often” to 7 = “very rarely/never”). A total score ranging from low to high levels of SOC was calculated [51,52]. The mean was reported in the analyses. The authors of the German version reported Cronbach’s alpha of 0.87 [51].

#### 2.4.3. Perceived Self Efficacy

The perceived self-efficacy of caregivers was assessed with a well-established questionnaire developed for teacher populations and slightly adapted by the authors for professional caregivers (4-point Likert scale with 10 items, scored from 1 = “not true” to 4 = “exactly true”) [53]. The meanwas reported in the analyses. The authors reported Cronbach’s alphas between 0.71 and 0.92 [53].

#### 2.4.4. Self-care Questionnaire

This author-developed questionnaire assessed physical, psychological and work-related self-caring behaviour [54]. The reference period reflected the past 3 months (4-point Likert scale with 24 items, scored from 1 = “completely inaccurate” to 4 = “completely accurate”). After conducting a principal components analysis to reduce data, three factors were extracted and rotated using promax-rotation (kappa = 4): (a) physical factors (e.g., participating in sports, sleeping enough, balancing nutrition), (b) psychological factors (e.g., feeling supported, upholding values, self-reflection) and (c) work-related factors (e.g., taking breaks, successfully transitioning from work to personal life, sharing responsibilities). In our sample, Cronbach’s alpha was 0.84. The selectivity of the items ranged from 0.22 to 0.59, while item difficulty ranged from 0.56 to 0.93. The total score mean was calculated for further analyses.

#### 2.4.5. Survey about Work-related and Personal Stressors

This author-developed survey documented the presence of work-related stressors in youth residential care, as well as typical personal stressors for adults [55]. Participants answered “yes” or “no” from a list of specific stressors experienced in the last three months prior to questioning. The 19 items on work-related stressors included exposure to aggression from clients (e.g., insults, kicked, spat on), aggression among children and adolescents (e.g., fighting) and self-injuring or suicidal behaviour of clients [10]. The 15 items on personal stressors documented life events such as divorce, severe accident or physical illness, moving, death of a loved one or birth of a child, including an open question to give participants the opportunity to address further stressors. Due to the confounding potential on burnout symptoms, the sum totals of reported work-related and personal stressors were controlled for during the statistical analyses [1,56,57,58].

### 2.5. Statistical Method

We explored associations between sociodemographic data and resilience measures by calculating analyses of variance (ANOVA) for categorical variables and Pearson’s correlation for continuous variables. Since some authors have argued that resilience is a holistic tendency and not concept specific, we performed bivariate Pearson’s correlation to test for associations between the SOC, self-efficacy and self-care constructs [22,59,60]. We calculated Pearson’s correlations to test associations between burnout and resilience measures at study entry. We calculated logistic regression models to determine the odds of burnout at study entry in relation to SOC, self-efficacy and self-care in independent models and then in a combined model. Last, we calculated Cox proportional hazards regression models to test the longitudinal association between SOC, self-efficacy and self-care and the risk of developing burnout during the course of the study. The Cox proportional hazards regression is sensible for analysing continuous-time event occurrence data [61]. It allowed us to examine and compare estimates for time-varying predictors, while also taking individual temporal modelling and differing number of measurement occasions across participants into account. We calculated models for each predictor independently and then combined. The Cox model time scale represented the time in months from the initial measurement point until onset of burnout or the last measurement point at which no burnout was reported (censoring). All cox models initially included the covariates sex, age, work experience in youth residential care, and employment years in the current institution, but none of these variables were significant predictors of time to burnout and therefore subsequently dropped from the models. All logistic regression and cox models were based on 2000 bootstrap samples. Statistical analyses were conducted using IBM SPSS (version 25, SPSS Inc., Chicago, IL, USA). All analyses were two-sided with the alpha level set at 0.05.

## 3. Results

### 3.1. Sociodemographic Variables

Associations between sociodemographic variables, reported personal and work-related stressors and resilience measures were analysed for all included participants at study entry (Table 1). Male participants reported higher values in self-efficacy compared to female participants. Older participants and those with children reported higher SOC and self-efficacy scores. The number of work-related stressors was negatively associated with self-efficacy and self-care. Being in a stable relationship, employment years in the current institution, work experience and personal stressors were not related to any resilience measures.

### 3.2. Bivariate Correlations between SOC, Self-efficacy and Self-care at Study Entry

Bivariate Pearson’s correlations were conducted to analyse associations between the SOC, self-efficacy and self-care constructs at study entry. The three resilience measures were found to be highly correlated with each other (Table 2).

### 3.3. Association between Resilience Measures and Burnout at Study Entry

Pearson’s correlation was analysed to test for associations between burnout symptoms in different life domains and resilience measures at study entry (Table 3). Difficulties in work-related, personal and interpersonal domains were negatively associated with SOC and self-care. Self-efficacy was only linked to difficulties in work-related and personal, but not interpersonal domains. Initial cognitive symptoms (e.g., “I have difficulties concentrating”, “I often perceive things negatively”) were negatively associated with SOC, self-efficacy and self-care, however somatic symptoms (e.g., “I have difficulty sleeping, “I suffer from headaches”) and emotional symptoms (e.g., “I feel anxious”, “I am irritable and tense”) were only associated with SOC and self-care.

At study entry, 31 participants were considered at-risk for burnout compared to 128 participants not at-risk. We calculated three logistic regression models to assess the cross-sectional association between SOC, self-efficacy, self-care and burnout. SOC (OR = 0.40, 95% CI [−1.46, −0.59], *p* < 0.001), self-efficacy (OR = 0.53, 95% CI [−1.12, −0.21], *p* = 0.003) and self-care (OR = 0.34, 95% CI [−1.76, −0.62], *p* < 0.001) were associated with burnout. Scoring one standard deviation above the mean in any resilience measure reduced the odds of being at risk for burnout at study entry.

In a second step, we calculated a combined model with all three predictors, which adjusted for associations between the resilience measures. Only SOC (OR = 0.50, 95% CI [−1.33, −0.26], *p* = 0.004) and self-care (OR = 0.42, 95% CI [−1.66, −0.30], *p* = 0.001) predicted burnout, whereas self-efficacy (OR = 0.82, 95% CI [−0.77, 0.38], *p* = 0.437) did not.

### 3.4. Longitudinal Analysis of the Relative Burnout Risk

To analyse the longitudinal association between resilience measures and burnout risk, participants reporting no burnout at study entry (*N* = 128) were investigated. Excluded from the analysis were 19 participants who did not have data for at least two consecutive time-points. Of the 109 remaining participants, 40 participants (36.7%) developed burnout during the course of the study. We calculated multiple Cox regression models, estimating the time to the incidence of burnout predicted by SOC, self-efficacy and self-care, first independently for each predictor and then combined in one model. In separate models, all three resilience measures were associated with reduced burnout risk (SOC: HR = 0.45, 95% CI [−1.18, −0.49], *p* < 0.001; self-efficacy: HR = 0.61, 95% CI [−0.87, −0.19], *p* = 0.003; self-care: HR = 0.68, 95% CI [−1.12, −0.20], *p* = 0.012). In a second step, we calculated a combined model that predicted the time to the development of burnout including all three predictors. In this model, only SOC and self-care were significantly associated with reduced burnout risk (Table 4).

## 4. Discussion

In this prospective study, we investigated the longitudinal association between specific resilience measures and burnout in a Swiss population of professional caregivers working in youth residential care. To our knowledge, this is the first longitudinal analysis of the association between burnout, SOC, self-efficacy and self-caring behaviour in the setting of youth residential care.

In line with previous research, SOC, self-efficacy and self-care were related to lower burnout symptoms in work-related and personal domains [19,21,32,35,36]. This was especially the case for cognitive symptoms, such as perceiving things negatively and reacting instead of acting. At study entry, individuals reporting higher values in all three resilience measures, especially SOC and self-care, had lower odds for burnout.

Our results indicate sociodemographic differences in individual resilience. Both SOC and perceived self-efficacy were positively associated with sex (identifying as male), older age and having children. We found no link to relationship status or years of work experience, but SOC was positively associated with number of employment years. Despite our findings, reports on sex differences vary and it remains an unstable predictor for burnout or resilience [25,41,42]. Older age, on the other hand, has been reported as a strong predictor for greater resilience such as SOC and lower burnout levels [24,38,40,62]. Although previous studies were not all in agreement, older caregivers may have developed stronger beliefs in their own ability to deal with stressful situations with growing experience and years of employment, using their available resources more effectively [27]. Alternatively, a survival bias may result in more resilient caregivers ‘surviving’ for longer, while those who are less resilient leave earlier [23,42]. The association between resilience, having children and employment years may indirectly tie in with older, more resilient individuals also being more likely to have children or be long-term employees [44]. Critical life events may change a person’s world views and affect SOC [63,64]. It seems plausible that as a life-changing event, having children could influence such perceptions favourably. Studies have reported that spending time with loved ones and developing meaningful relational roles outside of work have a protective effect [24,44,65]. Parents may also practice stricter work-life balance, which is beneficial for coping with job demands. Contrary to a recent study on self-care practices of child welfare workers, we found no sociodemographic differences in self-care [22]. Alkema et al. presumed that unrelated to age, professionals who take better care of themselves in various areas are less likely to leave the profession early due to burnout [23]. Thus, all individuals, no matter age or experience, are susceptible if they do not care for themselves.

During the course of our study, higher resilience scores reduced the risk of burnout. When all three measures were compared in the same statistical model, we found SOC and self-care to have the strongest protective effect. Our findings corroborate with an analysis on risk and protective factors in nursing, where no direct association between self-efficacy and burnout was found [41]. The SOC and self-efficacy constructs overlap in many respects, both including a cognitive component enabling the anticipation of events, a motivational component determining goal setting and a personal investment and a capability component, i.e., belief in one’s coping abilities [24]. This may explain the non-significant effect of self-efficacy when including all three resilience measures in the same model.

SOC was neither associated with the number of personal nor work-related stressors, suggesting SOC has little to do with whether stressors are perceived, but rather determines how they are coped with, i.e., are they still manageable, meaningful and comprehensible? A unique component of SOC is the aspect of ‘meaningfulness’, where there are no outcome expectancies and life events are understood as challenges rather than burdens [20]. Trap et al. found that individuals with low SOC are most likely to increase their SOC level [24]. As suggested by Vinje et al. in their professional training program, health promotion practices should therefore focus on expanding capacities in such individuals [66]. Interventions focusing on coping, problem-solving, cognitive therapy or lifestyle changes have been reported as effective [19].

Our findings on self-caring practices of professional caregivers support their relevance as a buffer in stressful work environments. Supporting our findings, Salloum et al. found that addressing self-care needs that are relevant for working with traumatized clients, such as trauma-informed self-care, which includes seeking supervision, working within teams, balancing caseloads and developing a plan for work-life balance, is protective against the development of burnout [21]. Furthermore, some authors have suggested that taking part in a variety of self-care strategies, not just one or two, may be more effective in managing symptoms [10,22]. Nevertheless, a recent systematic review concluded that self-care still takes a back seat in social work, and little is known about the efficacy of specific self-care practices [22,67]. More intervention research and integration into educational programs is needed for improving self-care competency and maintaining an empowered and healthy workforce.

The present study has certain limitations. Since reports on burnout and resilience were based solely on self-reports, a certain report/recall bias should be considered. However, the advantage of this is that inter-individual differences in stress perception are taken into account. With varying individual time intervals and measurements, we did not have four analogous cross-sectional measuring points to determine if sociodemographic results and correlations described at study entry remained consistent throughout the study period. Nevertheless, we were able to take this into account in the longitudinal analyses. Due to the high care standards found in institutions approved by the Swiss Federal Office of Justice, generalizations on an international scope should be made cautiously. The selection bias of participating youth welfare institutions may contribute to underestimations of burnout and overestimations of resilience. Unsurprisingly, we found SOC, self-efficacy and self-care to be highly correlated with one another, making individual interpretations more difficult. Some authors have suggested resilience to be a holistic tendency rather than being concept specific, so analysing the resilience measures in separate models as well as in the same model took this into account [22,59,60]. Furthermore, we did not systematically control for team dynamics and institutional problems, which may also have a relevant impact on work satisfaction and burnout.

## 5. Conclusions

Youth welfare organizations would benefit from future research assessing the effectiveness of professional and educational training programs focused on enhancing SOC and self-care practices. In the interest of cultivating a stable work environment, not only employees, but also leadership styles, case reviews and supervision should encourage staff engagement and self-care. In particular, younger employees just starting off in their careers and individuals perceiving work stressors as uncontrollable, meaningless and overwhelming could benefit in regard to performance, satisfaction, organizational commitment, as well as opting to stay on. We, however, also argue that the self-optimization of employees has its limits and is no substitution for organizationally embedded solutions to optimizing work environments. When drawing comparisons to the general population, many professional caregivers remaining on the job demonstrate above-average coping capacities. It is the duty of occupational policies to ensure that individuals with average, adequate health practices can enjoy their profession and continue working long-term with the vulnerable clients in their care.

## Figures and Tables

**Table 1 ijerph-17-02212-t001:** Cross-sectional analysis of sociodemographic variables and resilience measures at study entry for the study population of professional caregivers.

*N* = 159	Sense of Coherence		Self-Efficacy		Self-Care	
	M (SD)	*p*	M (SD)	*p*	M (SD)	*p*
Sex ^a^						
Male (*N* = 67) Female (*N* = 92)	5.76 (0.68) 5.54 (0.73)	0.048 *	3.17 (0.28) 3.06 (0.29)	0.013 *	3.25 (0.30) 3.32 (0.31)	0.156
Stable relationship ^a^						
Yes (*N* = 115) No (*N* = 33)	5.69 (0.71) 5.42 (0.69)	0.065	3.13 (0.28) 3.07 (0.32)	0.250	3.30 (0.31) 3.30 (0.29)	0.942
Own children ^a^						
Yes (*N* = 62) No (*N* = 97)	5.81 (0.70) 5.52 (0.71)	0.013 *	3.20 (0.28) 3.04 (0.28)	<0.001 ***	3.28 (0.34) 3.30 (0.29)	0.625
	r	*p*	r	*p*	r	*p*
Age ^b^	0.21	0.006 *	0.23	0.005 **	−0.02	0.776
Current empl. (yrs) ^c^	0.18	0.048 *	0.17	0.073	−0.12	0.212
Work exp. (yrs) ^c^	0.11	0.218	0.14	0.099	−0.12	0.156
Work-related stressors ^c^	−0.21	0.129	−0.16	0.042 *	−0.18	0.027 *
Personal stressors ^c^	−0.14	0.087	−0.03	0.737	−0.32	0.687

M = Mean, SD = Standard deviation, r = correlation coefficient; ^a^ ANOVA; ^b^ Pearson’s correlation; ^c^ Spearman’s correlation; * *p* < 0.05, ** *p* < 0.01, *** *p*< 0.001.

**Table 2 ijerph-17-02212-t002:** Bivariate Pearson’s correlations between sense of coherence, self-efficacy and self-care at study entry.

	Self-Efficacy		Self-Care	
	r	*p*	r	*p*
Sense of coherence	0.37	<0.001 ***	0.37	<0.001 ***
Self-efficacy	-		0.33	<0.001 ***

r = correlation coefficient; *** *p* < 0.001.

**Table 3 ijerph-17-02212-t003:** Associations between burnout symptoms in different life domains and sense of coherence, self-efficacy and self-care.

	Sense of Coherence		Self-Efficacy		Self-Care	
	r ^a^	*p*	r ^a^	*p*	r ^a^	*p*
**Domains**						
Work-related	−0.48	<0.001 ***	−0.22	0.005 **	−0.55	<0.001 ***
Personal	−0.47	<0.001 ***	−0.18	0.025 *	−0.57	<0.001 ***
Family	−0.28	<0.001 ***	−0.01	0.994	−0.34	<0.001 ***
Friend	−0.28	<0.001 ***	−0.06	0.443	−0.44	<0.001 ***
**Symptoms**						
Somatic	−0.34	<0.001 ***	−0.13	0.122	−0.45	<0.001 ***
Emotional	−0.53	<0.001 ***	−0.15	0.067	−0.45	<0.001 ***
Cognitive	−0.49	<0.001 ***	−0.20	0.013 *	−0.45	<0.001 ***

r = correlation coefficient; ^a^ Pearson’s partial correlation coefficients controlled for age and sex; * *p* < 0.05, ** *p* < 0.01, *** *p* < 0.001.

**Table 4 ijerph-17-02212-t004:** Longitudinal association between sense of coherence, self-efficacy, self-care and burnout risk during the course of the study in a combined cox regression model.

	HR ^a^	95% CI ^a^	*p* ^a^
Sense of coherence	0.58	[−1.04, −0.14]	0.004 **
Self-efficacy	0.77	[−0.61, 0.16]	0.112
Self-care	0.59	[−1.04, −0.22]	0.002 **

HR = Hazard Ratio, CI = Confidence Interval ^a^ CIs are based on 2000 bootstrap samples, ** *p* < 0.01.

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
