# Peer review of "What Protects Youth Residential Caregivers from Burning Out? A Longitudinal Analysis of Individual Resilience"

_ijerph, 2020, doi:10.3390/ijerph17072212_

Round 1
Reviewer 1 Report
- Abstract (line 14). Specify the study design (prospective longitudinal study).
- Study population. Sample estimation data (specifying confidence level, expected proportion, precision...) are missing. These are needed to determine the response rate achieved.
- Study population (line 83). It would be advisable to describe the main characteristics of the selected institutions (e.g. volume of population covered, user/professional ratio, age range of users, etc.)-
- Study population (lines 82-87). What was the professional profile of the caregivers, do all 160 participants belong to the same professional group?
- Procedures. Specify the range of years in which the field study was carried out.
- Measures. For the first four instruments, it is advisable to specify the type of Likert scale used (frequency, intensity, etc.) and the meaning of the extremes of the scale.
- Results. In the cross-sectional analyses of sections 3.1 and 3.2 it is not specified which of the four measures is being used. It should also be analysed and mentioned whether the results and correlations described are consistent across the four measures.
- Results, 3.3 Association between resilience measures and burnout at the study entry (lines 202-203). Review the interpretation of the significance of the odds ratio (OR) taking into account that null effect is 0. Therefore, those confidence intervals that include the value 0 indicate that there are no significant differences between the groups' odds. This is the case for both self-care and self-efficacy variables.
- Discussion (lines 231-232). It is recommended that an explanation be included, accompanied by appropriate references, regarding the association of SOC and self-efficacy with older age and having children.
Author Response
1. Abstract (line 14). Specify the study design (prospective longitudinal study).
Thank you very much for pointing this out. We have added the information as follows (line 14):
“Using a prospective longitudinal study design, participants (n=169; 57.9% women) reported on burnout symptoms and sense of coherence (SOC), self-efficacy and self-care at four annual sampling points.”
2. Study population. Sample estimation data (specifying confidence level, expected proportion, precision...) are missing. These are needed to determine the response rate achieved.
We thank the reviewer for declaring this. As mentioned in line 84, data stems from a larger government-funded model project examining the efficacy of trauma-informed care in residential youth welfare institutions. The sample size estimation is based on this primary study aim. This current secondary explorative analysis incorporated the already available data, so no sample estimation to determine response rate was conducted. We now explain this rationale in the Methods section (line 92-94):
“Sample size estimations were originally based on the primary study aim. Due to the secondary, explorative nature of the current analyses, no sample estimation to determine response rate was conducted.”
3. Study population (line 83). It would be advisable to describe the main characteristics of the selected institutions (e.g. volume of population covered, user/professional ratio, age range of users, etc.)
Thank you for this valuable feedback. We have taken the reviewer’s advice and described the main characteristics of the selected institutions (line 86-91):
“Included welfare institutions were approved by the Swiss Federal Office of Justice and incorporated a broad range of clients placed in out-of-home care for both civil reasons (e.g. family conflicts, neglect) and juvenile justice reasons (e.g. educational measures, reintegration). Six institutions were sex-specific (3 for boys, 3 for girls, ages 12-20 years old), while eight institutions were co-educative (ages 5-20 years old). The 14 youth welfare institutions offer placements for almost 300 clients.”
4. Study population (lines 82-87). What was the professional profile of the caregivers, do all 160 participants belong to the same professional group?
The reviewer asks an important question. All participants belong to the same professional group and are either social pedagogues or social pedagogues in training. We have specified this as follows (lines 97-98):
“Thus, 159 professional caregivers, i.e. social pedagogues or social pedagagues in training (67 men, 92 women) aged between 22 and 61 years (mean = 35.85, SD = 9.68) who worked in 14 residential youth welfare institutions were included in the study.”
5. Procedures. Specify the range of years in which the field study was carried out.
We appreciate this suggestion and have specified the range of years as follows (line 119-121):
“We used a longitudinal design over the course of three years to estimate changes in reported burnout of youth residential care staff over time. Surveys and well-established questionnaires were mailed to partaking institutions at four annual sampling points between 2012 to 2015.”
6. Measures. For the first four instruments, it is advisable to specify the type of Likert scale used (frequency, intensity, etc.) and the meaning of the extremes of the scale.
This is a very valuable suggestion. We have added the meaning of the extremes of the scales in the first four instruments. We believe this also provides sufficient information on the type of Likert scale used (line 135 – 154).
7. Results. In the cross-sectional analyses of sections 3.1 and 3.2 it is not specified which of the four measures is being used. It should also be analysed and mentioned whether the results and correlations described are consistent across the four measures.
Thank you, this is in need of clarification. The cross-sectional analyses were completed at study entry (measure 1). We specify this in section 3.1 and 3.2:
- “Associations between sociodemographic variables, reported personal and work-related stressors and resilience measures were analysed for all included participants at study entry (Table 1).” (line 207)
- “3.2 Bivariate correlations between SOC, self-efficacy and self-care at study entry” (line 220)
- “Bivariate Pearson’s correlations were conducted to analyse associations between the SOC, self-efficacy and self-care constructs at study entry.” (line 222)
The reviewer makes an important observation concerning correlation consistency across all four measures. Due to the continuous study inclusion, not all individuals have four measurements and interval times vary. This means that we unfortunately do not have four ‘clean’ analogous measuring points to conduct repeated cross-sectional analyses. Aware of this design challenge, we specifically opted for conducting longitudinal analyses with cox proportional hazards regression which accounts for inter- and intra-individual time variations between measures, as well as the varying measurement number (Singer & Willet, 2003). We have added this limitation to our manuscript (line 491 – 516):
“With varying individual time intervals and measurements, we did not have four analogous cross-sectional measuring points to determine if sociodemographic results and correlations described at study entry remained consistent throughout the study period. Nevertheless, we were able to take this into account in the longitudinal analyses.”
8. Results, 3.3 Association between resilience measures and burnout at the study entry (lines 202-203). Review the interpretation of the significance of the odds ratio (OR) taking into account that null effect is 0. Therefore, those confidence intervals that include the value 0 indicate that there are no significant differences between the groups' odds. This is the case for both self-care and self-efficacy variables.
We are very grateful that the reviewer noticed this error. The confidence interval reported for self-care was incorrect. We have corrected this oversight and apologize for the mistake (line 409):
“Only SOC (OR = .50, 95% CI [-1.33, -.26], p=.004) and self-care (OR=.42, 95% CI [-1.66, -.30], p=.001) predicted burnout, whereas self-efficacy (OR=.82, 95% CI [-.77, .38], p=.437) did not.”
9. Discussion (lines 231-232). It is recommended that an explanation be included, accompanied by appropriate references, regarding the association of SOC and self-efficacy with older age and having children.
We thank the reviewer for this suggestion. We have greatly expanded our discussion of sociodemographic variables and references to explain the association between SOC and self-efficacy with older age and having children (lines 438-461):
“Older age on the other hand is reported as a strong predictor for greater resilience such as SOC and lower burnout levels [24,38,40,63]. Although previous studies are not all in agreement, older caregivers may have developed stronger beliefs in their own ability to deal with stressful situations with growing experience and years of employment, using their available resources more effectively [27]. Alternatively, a survival bias may result in more resilient caregivers ‘surviving’ for longer, while those who are less resilient leaving earlier [23,42]. The association between resilience, having children and employment years may indirectly tie in with older, more resilient individuals also being more likely to have children or be long-term employees [44]. Critical life events may change a person’s world views and affect SOC [62,64]. It seems plausible that as a life-changing event, having children could influence such perceptions favorably. Studies report that spending time with loved ones and developing meaningful relational roles outside of work have a protective effect [24,44,65]. Parents may also practice stricter work-life-balance, which is beneficial for coping with job demands.”
Reviewer 2 Report
Kind et al.: What protects youth residential caregivers from burning out? A longitudinal analysis of individual resilience.
In this longitudinal study authors investigated the association between resilience and burnout in a Swiss population of professional caregivers working in youth residential care and found that especially SOC and self-caring behavior protect against burnout. The study is well designed, correctly conducted. Appropriate statistics were used. Results are moderately interpreted. Investigating the role of individual resilience on burnout is highly relevant for youth welfare organizations and occupational health policies thus this study is worth for the attention of the readers.
Detailed comments:
In the study population section, line 82, authors state that 160 caregivers were included in the study. But in the results section, in line 194, only 159 participants (31+128) are mentioned: ‘At study entry, 31 participants were considered at-risk for burnout compared to 128 participants not at-risk.’ Please clarify the reason of this discrepancy!
Author Response
In the study population section, line 82, authors state that 160 caregivers were included in the study. But in the results section, in line 194, only 159 participants (31+128) are mentioned: ‘At study entry, 31 participants were considered at-risk for burnout compared to 128 participants not at-risk.’ Please clarify the reason of this discrepancy!
We appreciate that the reviewer noticed this discrepancy. We reviewed our data and found that one participant had missing data in the burnout questionnaire which was overlooked. We apologize for this oversight and have now excluded this participant from the analysis due to 'missing data at study entry' and revised the sample size throughout the manuscript. The cross-sectional analyses were run again with the correct sample size (section 3.1 and 3.2). Two results which were barely not significant previously, are now significant – sex (identifying as male) and length of current employment were positively associated with sense of coherence. We discuss this in the Discussion section (4.0).